# Functional Characterization of the *MYO6* Variant p.E60Q in Non-Syndromic Hearing Loss Patients

**DOI:** 10.3390/ijms23063369

**Published:** 2022-03-21

**Authors:** Moza Alkowari, Meritxell Espino-Guarch, Sahar Daas, Doua Abdelrahman, Waseem Hasan, Navaneethakrishnan Krishnamoorthy, Abbirami Sathappan, Patrick Sheehan, Nicholas Van Panhuys, Xavier Estivill

**Affiliations:** 1Research Division, Sidra Medicine, Doha 26999, Qatar; mespinoguarch@sidra.org (M.E.-G.); sdaas@sidra.org (S.D.); dabdelrahman@sidra.org (D.A.); whasan@sidra.org (W.H.); nkrishnamoorthy2@sidra.org (N.K.); asathappan@sidra.org (A.S.); nvanpanhuys@sidra.org (N.V.P.); estivill@me.com (X.E.); 2College of Health and Life Sciences, Hamad Bin Khalifa University, Doha 34110, Qatar; 3Otolaryngology Division, Sidra Medicine, Doha 26999, Qatar; psheehan@sidra.org; 4Research Department, Quantitative Genomics Medicine Laboratories (qGenomics), 08950 Barcelona, Spain

**Keywords:** sensorineural hearing loss, *MYO6*, whole-genome sequencing, zebrafish, hair cells

## Abstract

Hereditary hearing loss (HHL) is a common genetic disorder accounting for at least 60% of pre-lingual deafness in children, of which 70% is inherited in an autosomal recessive pattern. The long tradition of consanguinity among the Qatari population has increased the prevalence of HHL, which negatively impacts the quality of life. Here, we functionally validated the pathogenicity of the c.178G>C, p.E60Q mutation in the *MYO6* gene, which was detected previously in a Qatari HHL family, using cellular and animal models. In vitro analysis was conducted in HeLa cells transiently transfected with plasmids carrying *MYO6^W^*^T^ or *MYO6^p.E60Q^*, and a zebrafish model was generated to characterize the in vivo phenotype. Cells transfected with *MYO6^WT^* showed higher expression of MYO6 in the plasma membrane and increased ATPase activity. Modeling the human *MYO6* variants in zebrafish resulted in severe otic defects. At 72 h post-injection, *MYO6^p.E60Q^* embryos demonstrated alterations in the sizes of the saccule and utricle. Additionally, zebrafish with *MYO6^p.E60Q^* displayed super-coiled and bent hair bundles in otic hair cells when compared to control and *MYO6^WT^* embryos. In conclusion, our cellular and animal models add support to the in silico prediction that the p.E60Q missense variant is pathogenic and damaging to the protein. Since the c.178G>C *MYO6* variant has a 0.5% allele frequency in the Qatari population, about 400 times higher than in other populations, it could contribute to explaining the high prevalence of hearing impairment in Qatar.

## 1. Introduction

Autosomal recessive non-syndromic hearing loss (ARNSHL) accounts for more than 70% of hereditary deafness, with risk alleles traced to more than 65 loci in the human genome, including multiple sites of myosin-encoding genes [1]. Myosins are a superfamily of proteins that bind to actin and hydrolyze ATP for energy production. Myosin proteins are commonly composed of head, neck, and tail domains, which bind to actin and generate movement via the catalytic activity of the head domain. Members of the myosin superfamily follow a unified mechanoenzymatic cycle that utilizes hydrolyzed ATP for movement along actin filaments [2,3,4]. Briefly, the mechanoenzymatic cycle starts with the strong binding of actin to myosin to form the actomyosin complex in the absence of ATP. Once the ATP binds to myosin, this causes the dissociation of actin and the hydrolysis of ATP to ADP and inorganic phosphate (Pi). Then, actin rebinds to myosin in a weak binding state that causes a mechanical interaction. Later, the hydrolysis products ADP and Pi are released, resulting in conformational changes in the motor domain, which causes myosin to move over the actin filament [5].

The gene encoding the MYO6 protein is located in the long arm of chromosome 6 (6q14.1) and consists of 35 exons. Mutations in *MYO6* are associated with autosomal recessive (DFNB37) and autosomal dominant (DFNA22) hearing loss [6,7]. More than 19,000 variants have been reported in *MYO6*; however, only 79 variants are thought to be pathogenic and associated with deafness or other phenotypes (Deafness Variation Database (http://deafnessvariationdatabase.org/) (accessed on 3 January 2022). Myosin 6 (MYO6) is expressed in the inner hair cells of the cochlea and is required to maintain its normal structure and function [8,9]. MYO6 is a member of unconventional myosins, which are known to be associated with non-syndromic hearing loss (NSHL) [6,7,10,11,12]. Strong evidence has shown that mutations in *MYO6* are responsible for causing both autosomal recessive (DFNB37) and autosomal dominant (DFNA22) forms of NSHL [6,7]. Currently, the functional characterization of hearing loss mutations is studied in several different animal models, including Drosophila, mouse, and zebrafish [13,14,15]. The association of Myo6 mutations with hearing loss was first characterized in Snell’s waltzer mice [16], where defects in hearing, as well as some abnormalities in the vascular endothelial cells of the heart and lung, were detected [14,17].

Since the early 1980s, zebrafish have been described as an excellent model to study the developmental and genetic features of vertebrates, including the auditory system [18]. The complete genome sequencing of zebrafish revealed that 71% of human genes have at least one zebrafish ortholog [19], and another highly analogous shared characteristic is the similarity of the auditory system between humans and zebrafish.

Several previous studies have sought to characterize the role of *MYO6* in zebrafish and its association with different phenotypes, including hair cell development and arterial morphogenesis [15,20,21]. Because the inner ear of the zebrafish closely resembles the human ear structure, zebrafish have become an ideal model to study the development and pathology of the human inner ear [22,23]. The zebrafish inner ear consists of three semicircular canals that contain one sensory patch in each canal called a crista. The cristae consist of both sensory epithelial hair cells and supporting cells. Zebrafish inner ears also contain two additional sensory patches known as maculae and an associated stone-like structure known as the otolith. These sensory patches function as detectors for both motion and sound. The lateral line is another sensory organ that contains mechanosensory hair cells called neuromasts, which are analogous in development and differentiation to the hair cells of the inner ear [24]. The inner ear is used for the detection of sound and motion, whilst the lateral line senses water flow over the body’s surface. 

In our previous study, we identified several novel variants in patients diagnosed with autosomal recessive NSHL [25]. These included the missense variant c.178G>C in the *MYO6* gene, which was identified in two affected siblings from a consanguineous Qatari family. Currently, very few studies have sought to evaluate the pathogenicity of missense variants identified in the *MYO6* gene and their association with hearing loss [8,26,27]. Therefore, in order to validate the predicted pathogenicity of the c.178G>C missense variant, we utilized several different functional validation strategies, including in vitro, in silico, and zebrafish models, to characterize in detail the missense variant c.178G>C in the *MYO6* gene and its downstream effects on auditory pathology.

## 2. Results

### 2.1. Prevalence of the Novel MYO6 Variant p.E60Q in Qatari Population

As we described previously [25], the novel *MYO6* variant is predicted to be pathogenic and is located near the ATP binding site in the motor domain of the MYO6 protein (Table 1 and Figure 1B,C). To assess the prevalence of the novel variant p.E60Q within the Qatari population, we used the genomic sequence of the participants from the Qatar Genome Program (QGP) dataset [28]. The QGP is a population-level genome project that aims to sequence all Qatari nationals, with the goal of developing personalized medicine for the prevention, diagnosis, and treatment of diseases.

The allele frequency of the c.178G>C variant was determined to be 0.5% in the Qatari population, which is significantly higher than any other population previously studied (Table 1). All identified carriers in the QGP dataset of the homozygous missense p.E60Q variant were subsequently validated by Sanger sequencing (Figure 1A). The phenotype–genotype correlation shows that <50% of the participants carrying the p.E60Q variant self-reported that they suffer from hearing difficulties that require the use of a hearing aid.

### 2.2. MYO6 Variant Is Predicted to Reduce ATPase Activity by In Silico Protein Modeling

To investigate the impact of the p.E60Q variant on protein structure, we produced a molecular model of the motor domain region of MYO6 (Figure 2A). The motor domain of human MYO6 showed that the structure is mainly constructed of α-helices. The core region of the myosin VI motor provides the site for ATP/ADP binding (Y87, P99, N98, E159, F163, and Y107). This site was mapped using experimental evidence of myosin VI motor structures (PDB ID: 2V26, 2VB6, and 2X51), which suggest that a conformational adjustment in the motor domain is important for actin detachment and the hydrolysis of ATP. Notably, the location of the p.E60Q mutation is in the vicinity of the ATP site of *MYO6*. As such, p.E60Q is predicted to change the negatively charged side chain residue glutamic acid (E) to the polar residue glutamine (Q) with an amine group at the end, which can impact the interaction network and thus affect local stability. The identified key contact residues (T88, Y89, and P125) also appear to act as an interaction bridge between the mutation site and the nucleotide binding site. 

We hypothesized that the p.E60Q mutation indirectly affects the ATP binding site via the inter-linking residues and thus alters the ATPase activity. To confirm this hypothesis, an ATPase activity assay was performed using protein extracts from transfected HeLa cells with human wild-type (*MYO6^WT^*) and mutated (*MYO6^pE60Q^*) *MYO6*. The *MYO6^p.E60Q^*-transfected cells showed a reduction in ATPase activity when compared to the *MYO6^WT^*-transfected cells (*p* < 0.005) (Figure 2B). This finding supports our molecular modeling results, which indicated that the missense variant would affect the ATP binding site of the MYO6 protein.

### 2.3. MYO6^p.E60Q^ Variant Alters Cellular Protein Trafficking

MYO6 is a cellular protein predominantly expressed in the plasma membrane of most cell types [31]. Therefore, to investigate the effect of the missense variant (p.E60Q) on protein trafficking, we utilized plasmids carrying either human *MYO6^WT^* or *MYO6^p.E60Q^* to overexpress the proteins in the HeLa cell line. Western blotting of protein extractions from whole-cell lysate and cell membrane enrichment was performed to confirm the overexpression and localization of MYO6 in the cellular membrane (Figure 3). When comparing the MYO6 expression ratio of the membrane protein over the whole-lysate protein, *MYO6^p.E60Q^*-transfected cells showed a significant reduction in MYO6 protein compared to *MYO6^WT^*-transfected cells (Figure 3A,B). To determine whether there were any protein trafficking defects, immunofluorescent co-localization of MYO6 with pan-cadherin, which is a membrane and nuclear marker, was performed (Figure 3C–E). The relative intensity of MYO6 in both transfected cells (*MYO6^WT^* and *MYO6^p.E60Q^*) was significantly higher compared to control cells (Figure 3D), which confirms the overexpression of MYO6. In comparison, the ratio of MYO6/pan-cadherin was significantly higher in *MYO6^WT^* compared to the control group, whereas there was no significant difference in *MYO6^p.E60Q^* levels (Figure 3E). Together, these results demonstrate that the *MYO6^p.E60Q^* variant leads to a significant alteration in the cellular protein trafficking and localization of MYO6.

### 2.4. Functional Validation of MYO6 Variant in the Zebrafish Model

To investigate the effect of the human *MYO6^p.E60Q^* variant in zebrafish, we analyzed both morphological and behavioral phenotypes in the zebrafish model system. We generated a null zebrafish (ZF_myo6a_MO) model and rescued it with co-injections of synthetic human *MYO6* RNA of the wild type (*MYO6^WT^*) and the variant (*MYO6^p.E60Q^*). The survival and hatching rates were measured at 24 hpf and 72 hpf, respectively (Appendix A). Although no significant differences in the survival rate were observed between the different groups, there was a delay in the hatching rate of the injected groups when compared to the uninjected control group. After 72hpf, the development of the zebrafish inner ear was evaluated by assessing ear morphology, hair cell structure, and swimming behavior.

#### 2.4.1. Human *MYO6^p.E60Q^* Variant Results in Zebrafish Ear Morphological Defects and Abnormal Inner Ear Hair Cell Development

Measurement of the overall ear capsule perimeter showed no difference in ear size between the uninjected group and the group rescued with *MYO6^WT^* and *MYO6^p.E60Q^* (Figure 4A,D). On the other hand, *ZF_myo6a_MO* resulted in significantly reduced ear size compared to the *MYO6^WT^*-injected embryos and a reduction in the size of the two otoliths, the saccule and utricle (Figure 4D), which are responsible for hearing and balance, respectively [32]. *MYO6^p.E60Q^* and *ZF_myo6a_MO* showed a significantly smaller anterior utricle area in comparison to the *MYO6^WT^* group (Figure 4D). Additionally, the posterior saccular area was reduced significantly in both *MYO6^p.E60Q^* and *ZF_myo6a_MO* in comparison with *MYO6^WT^*.

To examine the hair cell morphology and abundance, we stained the injected 72 hpf zebrafish with fluorescent acetylated tubulin to visualize the hair bundles (Figure 5). While the hair bundles of the control embryos exhibited a long and straight appearance (Figure 5A), the hair cells of *ZF_myo6a_MO* embryos appeared bent, coiled, and shortened (Figure 5B). Additionally, the hair cells of the *MYO6^p.E60Q^*-injected zebrafish were less abundant and super-coiled compared to the hair cells rescued with the *MYO6^WT^* injection (Figure 5C,D). 

#### 2.4.2. Human *MYO6^p.E60Q^* Variant Resulted in Altered Zebrafish Auditory Responses

As *myo6a* in zebrafish is expressed in the hair cells of the inner ear and lateral line, we can assess the auditory-sensory behavioral response in zebrafish larvae [33]. Here, the *MYO6^p.E60Q^* embryos demonstrated less locomotive behavior in response to a tapping stimulus compared to both control and *MYO6^WT^* embryos (Figure 6A). In comparison, the *MYO6^WT^* zebrafish showed similar swimming activity to that in the control group. To rule out the effect of sensory stimuli on zebrafish movement, we measured the swimming activity in both light and dark conditions without tapping stimuli. No significant changes in swimming activity behavior were found between the different zebrafish groups examined (Figure 6B).

## 3. Discussion

In this study, we identified the carrier frequency and functionally characterized the pathogenicity of a c.178G>C, pE60Q *MYO6* variant that was previously identified in one Qatari family [25]. Upon screening for the variant in the Qatari population, we found the carrier frequency of the variant to be 0.5% in the population, which was determined to be significantly higher than that present in other population-level genetic screening studies [29,30].

The variant p.E60Q is located in the motor domain of the protein near the ATP binding site, and our in silico modeling suggested mutation-dependent changes in ATPase activity (Figure 2A). Our study showed that the p.E60Q variant affects protein trafficking to the plasmatic membrane and reduces ATPase activity (Figure 2B), suggesting the presence of a dysfunctional MYO6 protein that might affect anchoring or walking across actin filaments, causing impaired hair cell function. This result is concordant with the findings of the studies by Herzano et al. [8], where the p.D179Y variant in a mouse model, located in the same motor region as p.E60Q, also led to a decrease in steady-state ATPase rates.

Here, the phenotypic effect of the human missense variant c.178G>C *MYO6* in zebrafish embryos showed auditory dysfunction, including otic morphological defects and decreased and coiled inner ear hair, in addition to a decreased auditory response in swimming behavior. The analysis of ear and otolith morphology is a distinct characterization method that is used to screen for any otic defects in zebrafish [34]. In this study, the human missense variant c.178G>C caused a reduction in the size of the zebrafish saccule when compared to the size in *MYO6^W^*^T^ and control groups (Figure 4). Another important finding was the observation that the hair cells in the inner ear of the mutant embryos were super-coiled when compared to those of the *MYO6^WT^*-injected zebrafish. These findings are similar to the results of Seiler et al., who concluded that mutations in *MYO6* caused irregular and disorganized hair bundles [15]. Interestingly, the auditory response swimming behavior assay showed a remarkable defect in fish injected with *MYO6^p.E60Q^* in comparison with *MYO6^WT^* and control embryos (Figure 6).

In conclusion, the emergence and application of tools associated with whole-genome sequencing have led to swift advances in our ability to rapidly identify novel variants associated with hearing loss in patients. Here, we identified the variant c.178G>C in *MYO6* to be extremely frequent in the Qatari population, with a prevalence of 0.5%, and subsequently characterized the pathogenicity of the variant using in silico, in vitro, and an in vivo zebrafish model system by developing a streamlined approach with broad application for the further evaluation of additionally identified novel genetic variants affecting the auditory system. Using this system, it was determined that the pE60Q mutation affects ATP hydrolysis, which alters MYO6 movement across actin filaments, and triggers altered hair bundle development. Recently, several gene therapy trials have been conducted that target hair cells of the inner ear with the application of synthetic adeno-associated virus therapies, with the goal of increasing transduction efficiency and improving the auditory threshold in treated animals [35,36]. It is envisaged that strategies aimed at rapid whole-genome sequencing for the identification and characterization of pathogenic variants, as outlined in this study, in combination with advances in gene therapy approaches, will allow for the enhanced diagnosis and treatment of patients suffering from hearing loss in the near future, with the introduction of truly personalized approaches to therapy.

## 4. Materials and Methods

### 4.1. Sanger Sequencing

To verify the missense variant in the Qatari samples, we performed Sanger sequencing using Applied Biosystem following the manufacturer’s protocol. The following primers were used to amplify exon 3 of *MYO6*: 5′-TGCAACCAATTAAGCCCTTCTA-3′ and 5′TGCAAATGTGAGACAACATGGA-3′.

### 4.2. Cell Culture and In Vitro Analysis

HeLa cell lines were cultured in Dulbecco’s Modified Eagle’s Medium (DMEM)/high-glucose medium with 10% fetal bovine serum and supplemented with penicillin–streptomycin–glutamine. Cells were maintained in a humidified incubator at 37 °C and 5% CO_2_. Twenty-four hours before transfection, cells were trypsinized and seeded into a 6-well plate at a density of 3 × 10^5^ cells per well to reach 60–80% confluency. The lipofectamine 2000 reagent protocol (Invitrogen Life Technologies) was used for transfecting cells with 4 µg of DNA plasmid carrying wild type and missense variant. 

Total protein was extracted and loaded in SDS-PAGE gels as described previously, and the membrane was blotted using MYO6 MYO6 (M0691, Sigma-Aldrich, St. Louis, MO, USA) and (PA5-35054, Thermo Fisher Scientific, Waltham, MA, USA), and monoclonal anti-α-tubulin (T5168, Sigma-Aldrich) antibodies. 

ATPase assays were performed by using a malachite green-based colorimetric assay (MAK113, Sigma-Aldrich) according to the manufacturer’s protocol.

### 4.3. Immunofluorescent Staining, Confocal Microscopy Imaging, and Quantification

Confocal fluorescence images were acquired using the Zeiss LSM 880 Airyscan microscope. For different HeLa cell transfections, immunofluorescent staining was performed according to Abcam’s protocol. The antibodies used for the staining are: MYO6 (PA5-35054, Thermo Fisher Scientific), pan-cadherin (CH-19), (ab6528, Abcam, Cambridge, UK), and Alexa Flour secondary (A-11011 and A-11001 Thermo Fisher Scientific) antibodies. All prepared slides were analyzed at the same time using the same acquisition parameters. Quantification of the images was performed using ImageJ. 

### 4.4. Zebrafish Care and Husbandry

As described previously, adult wild-type (AB) zebrafish (Danio rerio) were maintained under standard environmental conditions: temperature of 28 °C, conductivity of 1000 µs, and pH of 7.0 for 14 h [37]. Zebrafish experiments were approved by the IACUC Office of Qatar University (QU-IACUC 26-2/2018-REN1).

### 4.5. Morpholino Design and Synthetic mRNA Injection

The myo6a morpholino (MO, 5′ CACCGGCTTTCCATCGTCCATTTCA 3′, Gene Tools, Philomath, OR, USA) targeted against the translational start site was injected into embryos at the one-cell stage to knock down endogenous zebrafish myo6a. MO antisense oligos were dissolved to a final concentration of 2.0 µM. Injections were performed at the one-cell stage using PLI-100 Picolitre injector, Harvard Apparatus, as described previously [37]. Embryos at the 1–2-cell stage were injected with 50 ng of ATG morpholinos to knock down endogenous zebrafish and to generate null zebrafish (ZF_myo6a_MO). The ZF_MYO6_MO model was rescued with co-injections of synthetic human *MYO6* RNA of the wild type (*MYO6^W^*^T^) and the variant (*MYO6^p.E60Q^*).

### 4.6. Head and Ear Imaging of Zebrafish Larvae 

Zebrafish head morphology was examined when they were 3 days old. Images of the head were captured by Zeiss Lumar 12 stereomicroscope. Zebrafish larvae were mounted in 3% methylcellulose for stabilization throughout imaging time. The ear, utricle, and saccule areas were measured using Danioscope software (version 1.1, Noldus, Wageningen, The Netherlands).

### 4.7. Zebrafish Staining and Imaging

For zebrafish imaging, fixed whole zebrafish larvae were incubated at 37 °C overnight with blocking and permeabilization solution (1% BSA, 1% FBS, and 0.3% TritonX in PBS). Later, the larvae were incubated in the dark at 4 °C overnight with 1:200 dilution of Alexa Fluor^®^ 488 acetylated α-tubulin antibody (sc23950, Santa Cruz Biotechnology, Santa Cruz, CA, USA) in a mix of 1% BSA and 1% FBS solution and DAPI. After raising with PBS, the larvae were washed 3 times with 15%, 50%, and 75% glycerol for 20 min each. Then, the larvae were mounted onto dishes with fine glass, and confocal images were taken using 100× water lenses on Zeiss LSM 880 microscope.

### 4.8. Zebrafish Locomotor Behavior Measurements

The auditory-sensory behavioral response in zebrafish larvae was examined by the assessment of locomotive behavior through the presentation of light–dark cycles over time intervals of 30 min and tapping, as described previously [38]. The total distance moved by zebrafish larvae was calculated using Ethovision software (Noldus, Wageningen, The Netherlands).

### 4.9. Protein Modeling

The 3D structural protein and the mutant were studied using molecular modeling. The protein coded by the *MYO6* gene was constructed from humans (MYO6 UniProt: Q9UM54). The protein has no available 3D structure from X-ray crystallography or/and NMR. Therefore, we used structural homologs of myosin 6 motors (PDB ID: 2BKI (Gallus gallus, 98% identity, Reference [39], and 2V26 (Sus scrofa, 98% identity) [40]) to build a human MYO6 3D model (residues 1–780). The quality of the modeled structure was examined as previously described [36,41], which indicates the biological relevance of the structure. The generated model of the wild type was subsequently used to create a mutant (p.E60Q in MYO6) in Discovery Studio (Accelrys Inc., San Diego, CA, USA), as explained previously [42,43], and we also mapped the functional sites (ATP/ADP binding site), including the key residues. The model representations were produced using pyMOL software (Schrödinger, NY, USA).

### 4.10. Statistical Analyses

All statistical analyses and graphs were performed using GraphPad Prism Software version 9.3.1 (GraphPad, San Diego, CA, USA). All data are presented as the mean ± SEM. 

## Figures and Tables

**Figure 1 ijms-23-03369-f001:**
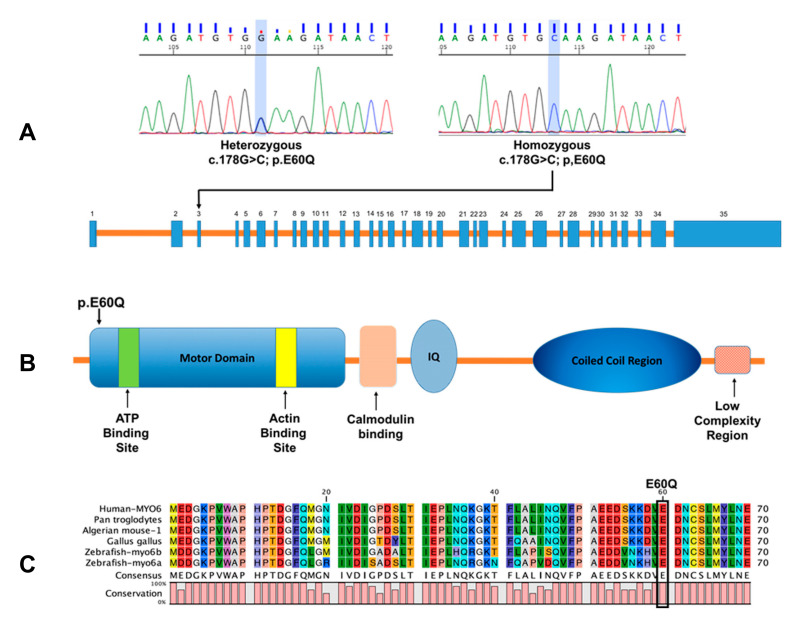
Mutation analysis of c.178G>C *MYO6* variant. (**A**) Sanger sequence of heterozygous and homozygous missense c.178G>C variant in exon 3 of MYO6. (**B**) Schematic diagram of myosin VI protein. (**C**) *MYO6* ontology across multiple species highlighting the conserved E60 residue.

**Figure 2 ijms-23-03369-f002:**
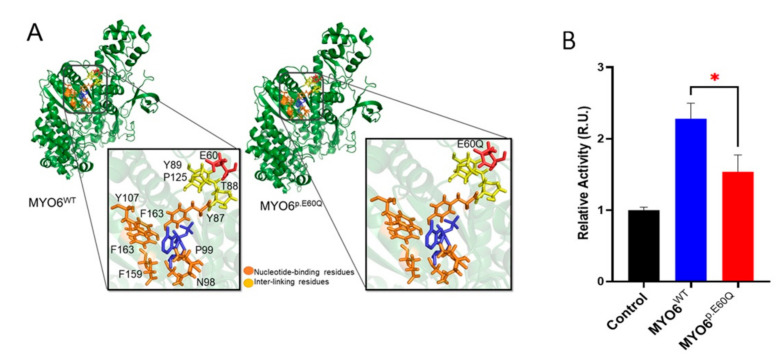
(**A**) Molecular protein structure of the motor domain of MYO6 showing E60 and the p.E60Q variant. (**B**) ATPase activity assay of whole-cell lysates from transfected HeLa cells with plasmids carrying human *MYO6^WT^* and *MYO6^p.E60Q^*. Enzyme activity was measured using a malachite green-based colorimetric assay. Values are represented as the mean ± SEM from independent experiments. Statistically significant differences were assessed by unpaired *t*-test, * *p* < 0.05.

**Figure 3 ijms-23-03369-f003:**
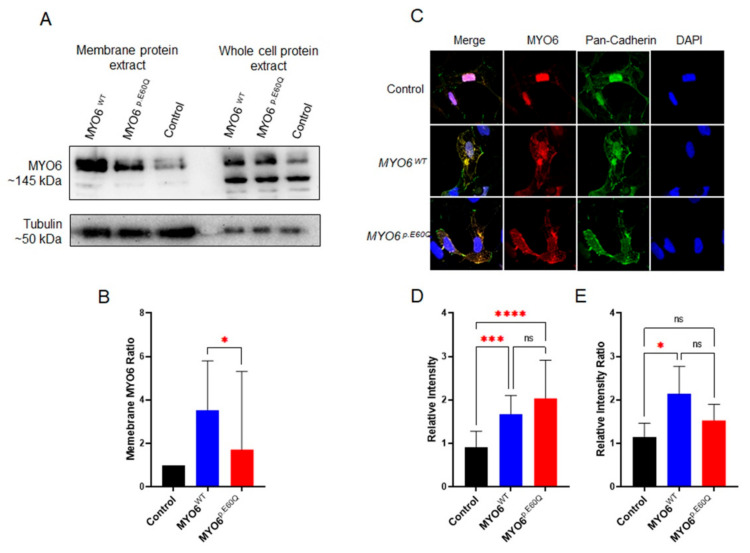
Cellular analysis of novel *MYO6* variant. (**A**) Western blot of whole-cell lysates and membrane proteins from *HeL*a cells transfected with plasmids carrying human *MYO6^WT^* and *MYO6^p.E60Q^.* (**B**) Quantification ratio of the normalized integrated density of MYO6 expression from membrane/whole proteins. (**C**) Representative immunofluorescence staining images of *HeLa* cells labeled by MYO6 (red), pan-cadherin (green), and DAPI (blue). (**D**) Quantification of the relative fluorescence intensity of MYO6. (**E**) Quantification of the ratio of the normalized relative fluorescence intensity of MYO6/pan-cadherin. All values are represented as the mean ± SEM from independent experiments. Statistically significant differences were assessed by (**B**) unpaired *t*-test, * *p* < 0.05 or (**D**,**E**) one-way ANOVA followed by Tukey’s multiple comparisons, * *p* < 0.05; *** *p* = 0.0001; **** *p* < 0.0001.

**Figure 4 ijms-23-03369-f004:**
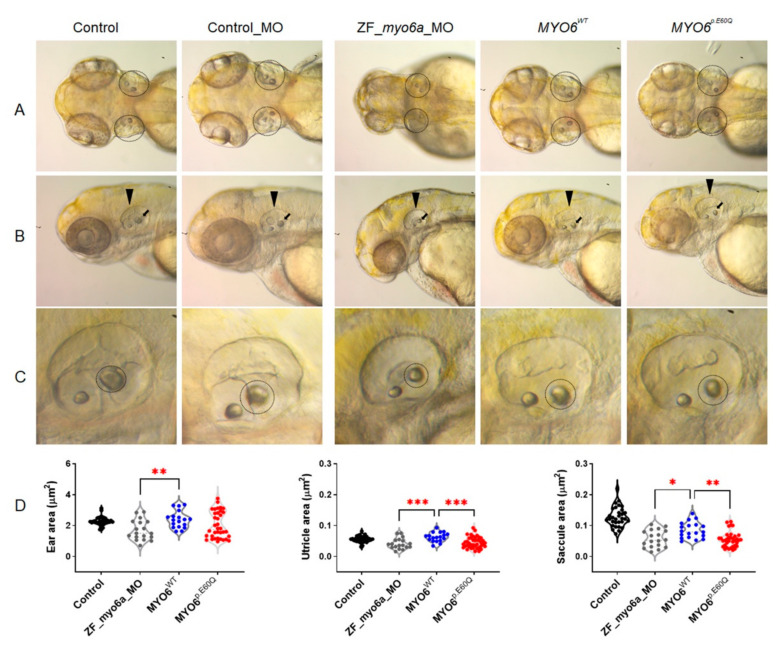
Detection of ear morphological defects in zebrafish models. (**A**) Dorsal view of zebrafish at 72 hpf. (**B**) Lateral view of the posterior otolith at 72 hpf zebrafish (black arrow). (**C**) Closeup view of zebrafish ear morphology (black dashed circle). (**D**) The perimeter length of zebrafish ear, utricle, and saccule area in μm^2^. The total number of experiments was 3, and the number of embryos analyzed was 31, 18, 19, and 36 for control, ZF_myo6a_MO, MYO6^WT^, and *MYO6^p.E60Q^* groups, respectively. One-way ANOVA using GraphPad Prism software (version 8.0) and Tukey’s multiple comparisons test with *p*-values of * *p* < 0.05, ** *p* < 0.01, *** *p* < 0.001.

**Figure 5 ijms-23-03369-f005:**
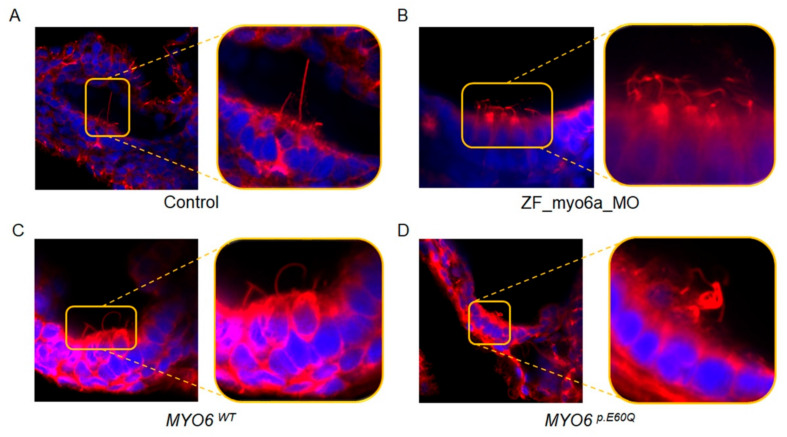
Inner ear hair cell characterization. Representative image of inner ear hair bundle phenotype of zebrafish larvae at 72 hpf of (**A**) Control, (**B**) *ZF_myo6a_MO* total knock-out, and the rescue with (**C**) wild type *MYO6^p.E60Q^* and (**D**) *MYO6^p.E60Q^* mutation. Representative image of inner ear hair bundle phenotype of zebrafish larvae at 72 hpf. F-actin-rich hair bundles were visualized using fluorescent acetylated tubulin and visualized using Airyscan confocal microscopy at a magnification of 100×.

**Figure 6 ijms-23-03369-f006:**
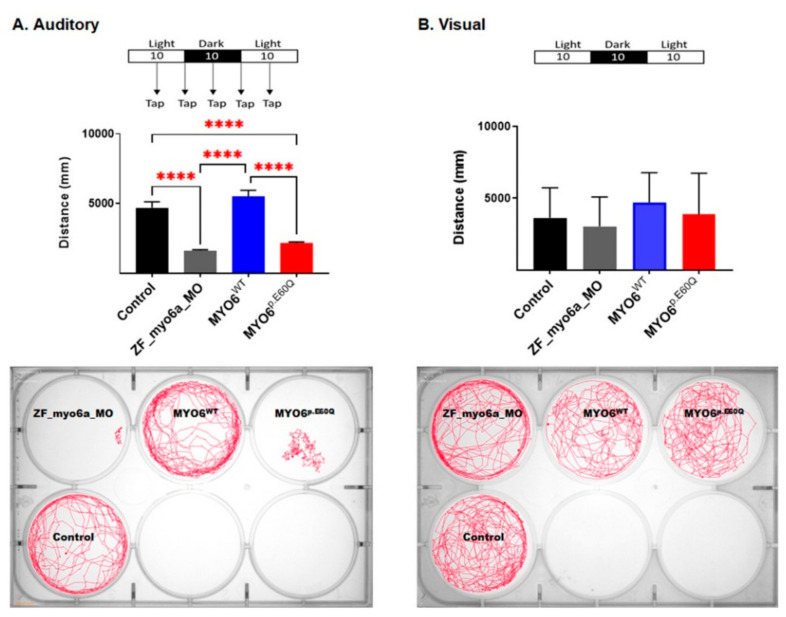
Auditory-sensory behavioral response in zebrafish larvae was examined by assessing locomotive behavior recorded through the presentation of light–dark cycles over time intervals of 30 min and tapping. The total distance moved by zebrafish larvae was calculated using Ethovision software (Noldus). (**A**). Auditory-sensory behavioral response. (**B**). Visual-sensory behavioral response. Representative behavior and response activity to auditory-sensory stimuli of zebrafish groups measuring movements over time for the examined zebrafish groups; movement is shown by the drawn red lines. The number of experiments *n* = 4; values represent mean with standard deviation. Statistical analysis was performed with one-way ANOVA followed by Tukey’s multiple tests for multiple comparisons, **** *p* < 0.0001.

**Table 1 ijms-23-03369-t001:** Pathogenicity and frequencies of p.E60Q *MYO6* variant in the Qatari population.

Mutation Prediction Tools		Allele Frequencies (%)	
MutationTaster ^1^	1	Qatar Genome Program	0.5
SIFT ^2^	0.01	GME ^7^	0.15
Polyphen-2 ^3^	0.994	genomAD—Exomes	0.0012
CADD ^4^	24.2	genomAD—Genomes [29]	0.0007
PhyloP ^5^	9.444	ExAC	0.0008
GERP++ ^6^	5.92	TopMed	0.0015

^1^ MutationTaster: closer to 1 is more likely to be damaging. ^2^ SIFT: closer to 0 is more damaging. ^3^ Polyphen-2: >0.85 is probably damaging, 0.85–0.15 is possibly damaging, and <0.15 is benign. ^4^ CADD: >10 is predicted to be deleterious. ^5^ PhyloP: >0.95 is conserved, and <0.95 is not conserved. ^6^ GERP++: >0 is generally conserved. ^7^ The Greater Middle East (GME) Variome Project [30].

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
