# Peer review of "Functional Characterization of the MYO6 Variant p.E60Q in Non-Syndromic Hearing Loss Patients"

_ijms, 2022, doi:10.3390/ijms23063369_

Round 1

Reviewer 1 Report

The paper is very large and difficult to read. It needs to be shortened.
The title does not reflect the content.  

The paper is not dedicated to patients and hearing impairment in patients with MYO6 gene mutation.
Keywords are also misleading.
The Information about the prevalence of the mutation in the population is redundant for this article, it belongs to another article.
Paper is oversaturated with informational material, which makes it difficult to read.
There are a lot of repetitions of the same text in the review and discussion, materials and methods.
We propose to combine information about the protein and zebrafish model in one place
Comments:
Р.2 - line 93  - this is a topic for a separate article.
P 5 line 196 - it is incorrect to mention the sacculus as the organ responsible for hearing. 
P 6 line 202 - the figure caption contains information on materials and methods
P 7 - line 230 - information on the zebrafish inner ear should be transferred to the review review.
P 8 - line 252 -258 - this is a review and replay of Р 2 line 94-107.
P 8 - line 259 - 269 - it should be in the review.

Reviewer 2 Report

Review on the manuscript

Functional Characterization of the MYO6 variant p.E60Q in non-syndromic hearing loss patients” by authors Moza Khalifa Alkowari * , Meritxell Espino-Guarch , Sahar Daas , Doua Abdelrahman , Waseem Hasan , Navaneethakrishnan Krishnamoorthy , Abbirami Sathappan , Patrick Sheehan , Nicholas Van Panhuys , Xavier Estivill

The authors present a functional characterization of the MYO6 variant p.E60Q founded early among patients with non-syndromic hearing loss in Qatar.

Major comments

No major comments, its good job. I do not see any major concerns.

Minor comments

  1. The long tradition of consanguinity among the Qatari population increases the prevalence of HHL, making it one of the most frequent causes of cognitive decline”. In my opinion, cognitive decline – it’s not the right term, I advise you to replace this term.
  2. Key words. I think there is a duplication – MYO6 and MYOVI.
  3. Throughout in text. The names of human genes should be written in capital letters, in italics; this is not observed everywhere (For example: Fig 1, Fig 5, M&M section); Please check this throughout.
  4. Throughout in text. In the text you write – “The allele frequency of the c.178G>C variant was determined to be 005 in the Qatari population”. However, in Table 1 allele frequency is presented in other in other values 0.5. In my opinion, it is necessary to bring the data in tables and in the text into uniform values.
  5. Figures. The text in all figures are difficult to read. Pleas enlarge the figures or text in all figures.
  6. Material and Method. The order of the items is broken. (For example the items 4.7. is duplicated). I think this section needs to be carefully checked.

Round 2

Reviewer 1 Report

Authors made corrections according comments and suggestions.